# Differentiation of Classical Swine Fever Virus Virulent and Vaccine Strains by CRISPR/Cas13a

Yuhang Zhang,[a] Qingmei Li,[b] Ruining Wang,[c] Li Wang,[b] Xun Wang,[a] Jun Luo,[b] Guangxu Xing,[b] Guanmin Zheng,[d] Bo Wan,[a] Junqing Guo,[b] 🔟Gaiping Zhang[a,b,e]

[a]International Joint Research Center of National Animal Immunology, College of Veterinary Medicine, Henan Agricultural University, Zhengzhou, China
[b]Key Laboratory of Animal Immunology, Henan Academy of Agricultural Sciences, Zhengzhou, China
[c]Henan University of Animal Husbandry and Economy, College of Veterinary Medicine, Zhengzhou, China
[d]Public Health and Preventive Medicine Teaching and Research Center, Henan University of Chinese Medicine, Zhengzhou, China
[e]Jiangsu Co-Innovation Center for Prevention and Control of Important Animal Infectious Disease and Zoonose, Yangzhou University, Yangzhou, China

**ABSTRACT** As a notifiable terrestrial and aquatic animal disease listed by World Organisation for Animal Health (formerly the Office International des Epizooties [OIE]), classical swine fever (CSF) has caused great economic losses to the swine industry worldwide during recent decades. Differentiation of infected and vaccinated animals (DIVA) is urgent for eradication of CSF. In this study, a diagnostic platform based on CRISPR/Cas13a was established with the ability to differentiate between classical swine fever virus (CSFV) virulent and vaccine strains. In combination with reverse transcription recombinase-aided amplification (RT-RAA), the detection limit for CSFV synthetic RNA templates reached $3.0 \times 10^2$ copies/$\mu$L. In addition, with boiling and chemical reduction, heating unextracted diagnostic samples to obliterate nucleases (HUDSON) treatment was introduced to inactivate nucleases and release viral genome, achieving robust pretreatment of tested sample before CRISPR/Cas13a detection without the need to extract viral nucleic acids. HUDSON-RT-RAA-CRISPR/Cas13a can directly detect cell cultures of virulent Shimen strain and vaccine hog cholera lapinized virus (HCLV) strain, with the detection limit of $3.5 \times 10^2$ copies/$\mu$L and $1.8 \times 10^2$ copies/$\mu$L, respectively, which was equally sensitive to nested PCR (nPCR) and 100 times more sensitive than antigen enzyme-linked immunosorbent assay (ELISA). Meanwhile, HUDSON-RT-RAA-CRISPR/Cas13a showed no cross-reactivity with bovine viral diarrhea virus (BVDV), atypical porcine pestivirus (APPV), porcine reproductive and respiratory syndrome virus (PRRSV), porcine epidemic diarrhea virus (PEDV), African swine fever virus (ASFV), pseudorabies virus (PRV), and porcine circovirus 2 (PCV2), exhibiting good specificity. At last, a total of 50 pig spleen samples with suspected clinical signs were also assayed with HUDSON-RT-RAA-CRISPR/Cas13a, nPCR, and antigen ELISA in parallel. HUDSON-RT-RAA-CRISPR/Cas13a showed 100.0% with nPCR and 82.0% coincident rate with antigen ELISA, respectively.

**IMPORTANCE** Classical swine fever (CSF) is a World Organisation for Animal Health (formerly the Office International des Epizooties [OIE]) notifiable terrestrial and aquatic animal disease, causing great economic losses to the swine industry worldwide during the past decades. Due to the use of the most effective and safe attenuated live vaccine for CSF prevention, differentiation of infected and vaccinated pigs is vital work, as well as a bottleneck for eradication of CSF. Methods with the ability to precisely differentiate classical swine fever virus (CSFV) virulent strains from vaccine strain hog cholera lapinized virus (HCLV) are urgently needed. Combining the high sensitivity of isothermal recombinase-aided amplification (RAA) with the accurate molecular sensing ability of Cas13a, we presented a novel method for CSFV detection without the need to extract viral nucleic acids, which showed great advantage to traditional detection methods for precise differentiation

**Ad Hoc Peer Reviewer** 🔟Hua-Ji Qiu, Harbin Veterinary Research Institute; 🔟Qin Zhao, Northwest A&F University

Address correspondence to Gaiping Zhang, zhanggaip@126.com, or Junqing Guo, 13838248132@163.com.

The authors declare no conflict of interest.

of CSFV virulent strains and vaccine strain, providing a novel powerful tool for CSF eradication.

**KEYWORDS** classical swine fever virus, recombinase-aided amplification, CRISPR/Cas13a, differentiation of infected and vaccinated animals

Classical swine fever (CSF) is a highly contagious and often fatal infectious porcine disease, leading to high fever, multisystemic hemorrhagic lesions, and immunosuppression in infected pigs. It has brought great economic losses to the swine industry worldwide during recent decades and is thus classified as one of notifiable terrestrial and aquatic animal diseases by the World Organisation for Animal Health (formerly the Office International des Epizooties [OIE]) (1).

Classical swine fever virus (CSFV), the causative agent of CSF, belongs to the family *Flaviviridae*, genus *Pestivirus*. It is a small enveloped virus with a single-stranded, positive-sense RNA genome. The genomic RNA contains a large open reading frame flanked with a 5′ untranslated region (5′UTR) and a 3′ untranslated region (3′UTR), encoding four structural proteins (C, $E^{rns}$, E1, and E2) and eight nonstructural proteins ($N^{pro}$, p7, NS2, NS3, NS4A, NS4B, NS5A, and NS5B) (2). Based on partial sequences of 5′UTR, E2 and NS5B, CSFV can be divided into three genotypes and 11 subgenotypes (1.1 to 1.4, 2.1 to 2.3, and 3.1 to 3.4) (3). Since the effective and safe attenuated live vaccine hog cholera lapinized virus (HCLV) strain was produced by serial passage of CSFV Shimen strain in rabbits at 1956, many countries, including China, use a systematic prophylactic vaccination policy to control or eradicate CSF, excepting for some European and American countries that carry out a stamping out policy (nonvaccination policy). To date, pandemics of CSF have been effectively prevented with either vaccination or nonvaccination policies. Instead, sporadic and occasional epidemics of large-scale outbreaks have become current epidemiological trend of CSF (4, 5).

Since 2017, Chinese government has begun to take a series of actions for CSF eradication, including vaccination and diagnostic policies. One obstacle of CSF elimination is that both the virulent Shimen strain and vaccine HCLV strain belong to the same serotype and subgenotpye 1.1, making it hard for traditional immunological or molecular diagnostic methods to differentiate infected pigs from vaccinated ones (6–10). Although efforts have been made to develop alternative subunit or gene-deleted vaccines for differentiation of infected and vaccinated animals (DIVA) purposes (11), HCLV-based attenuated live vaccine still remains the primary choice for CSF prevention in China. Methods with the ability to precisely differentiate CSFV virulent strains from vaccine strain HCLV are urgently needed. Current studies about CSFV DIVA mainly rely on molecular tests, including sequencing, PCR, and restrictive-fragment-length polymorphism (12, 13). However, these methods are often laborious and time consuming and need to be performed by skilled technicians with expensive instruments, using multiple testing steps and complex reagents.

Cas13a is a single-component RNA-guided RNA-targeting CRISPR effector, belonging to type VI-A within CRISPR/Cas family. Guided by CRISPR RNA (crRNA), Cas13a cleaves both the target RNAs and nonspecific single-stranded RNA (ssRNA) (14, 15). The nonspecific ssRNA cleavage activity of Cas13a is called collateral RNase activity and can be assayed with a fluorophore-quencher-labeled RNA substrate (reporter RNA) for fluorescent readout (16). In combination with recombinase-based isothermal amplification and collateral cleavage assay for Cas13a, a promising next-generation diagnostic strategy for infectious diseases was developed and named SHERLOCK for Specific High Sensitivity Enzymatic Reporter UnLOCKing. SHERLOCK is an isothermal method that can detect both DNA and RNA target at an attomolar level with single-base mismatch specificity (17).

In addition, Heating Unextracted Diagnostic Samples to Obliterate Nucleases (HUDSON) treatment can lyse viral particles and inactivate ribonucleases with the use of heat and chemical reduction, avoiding extraction of viral nucleic acids. With high performance and

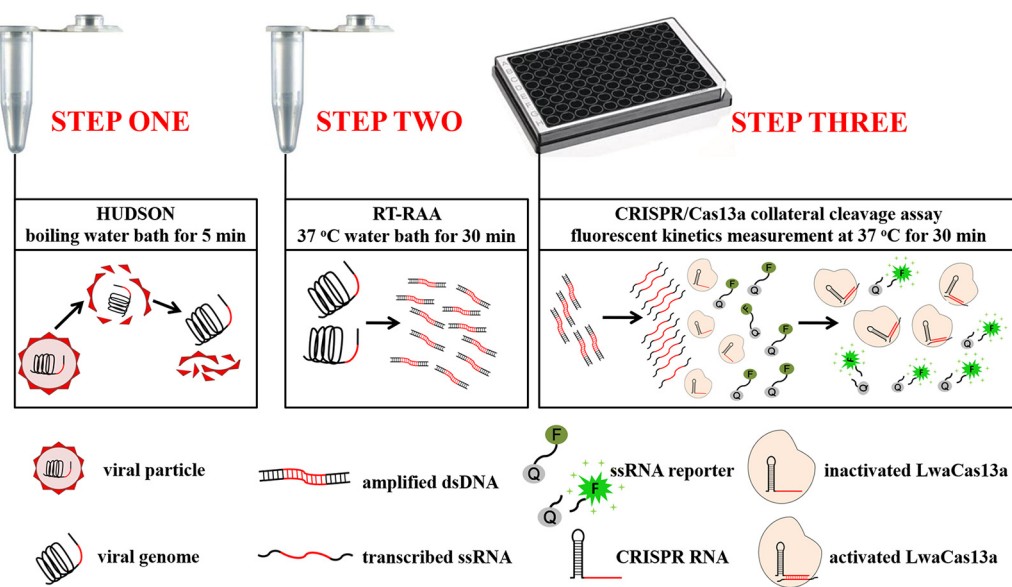

**FIG 1** Schematic of CRISPR/Cas13a diagnostic platform. The CRISPR/Cas13a diagnostic platform consists of three steps. Step 1 included HUDSON treatment of tested samples, which was performed at boiling water for 5 min to simultaneously inactivate nucleases and release viral genomic nucleic acids. Step 2 included RT-RAA of HUDSON-treated samples, which was performed at 37°C water bath for 30 min to isothermally amplify detecting target. Step 3 included CRISPR/Cas13a detection of RT-RAA products, which performed at 37°C for 30 min on the fluorescent reader. dsDNA, double-stranded DNA; HUDSON, heating unextracted diagnostic samples to obliterate nucleases; RT-RAA, reverse transcription recombinase-aided amplification; ssRNA, single-stranded RNA.

minimal sample processing requirements, HUDSON-SHERLOCK is a rapid and convenient diagnostic method that has been used for differentiation of four related *flaviviruses* viruses, including Zika virus, dengue virus, West Nile virus, and yellow fever virus (18).

Combined the high sensitivity of recombinase-aided amplification (RAA) with the accurate molecular sensing ability of Cas13a, in this study, we established a novel DIVA platform to differentiate between CSFV virulent strains and vaccine strain without the need to extract viral nucleic acids (Fig. 1), providing a promising strategy for CSF diagnosis and eradication.

## RESULTS

**Expression and purification of LwaCas13a.** A single band with a molecular weight of about 140 kDa was obtained by SDS-PAGE after nickel affinity chromatography and sulfopropyl (SP) cation exchange chromatography, illustrating that LwaCas13a was successfully expressed in soluble state and purified to a desired degree (Fig. S1). A total of 23.04 mg purified LwaCas13a was obtained from 3-liter bacterial LB cultures.

**CSFV crRNA screening.** Among five candidate crRNAs for HCLV strain, crHCLV5 produced the highest fluorescent signals, exhibiting the ability to recognize HCLV 3′UTR RNA and activate LwaCas13a efficiently (Fig. S2A). Similarly, among nine candidate crRNAs for the Shimen strain, crShimen7 produced the highest fluorescent signal, exhibiting the ability to recognize Shimen 3′UTR RNA and activate LwaCas13a efficiently (Fig. S2B).

**Estimation of the ability of CRISPR/Cas13a to recognize different CSFV genotypes with screened crRNAs.** To estimate the ability of crHCLV5 and crShimen7 to differentiate between CSFV vaccine strain and virulent strains, LwaCas13a collateral cleavage assay was performed to test different genotypes of CSFV 3′UTR RNAs, including subgenotypes 1.1, 1.2, 1,4, 2.2, 2.3, 3.2, and 3.4, as well as subsubgenotypes 2.1a, 2.1b, 2.1c, 2.1d, 2.1g, 2.1h, and 2.1i. For crHCLV5, only the cognate HCLV strain can be recognized, indicating that the vaccine strain can be differentiated from all tested virulent strains by CRISPR/Cas13a-crHCLV5 (Fig. 2A). For crShimen7, only four virulent subgenotypes can be recognized: 1.1, 1.2, 2.2, and 3.2 (Fig. 2B).

Considering the dominant pandemic subgenotype in China was 2.1, protospacer

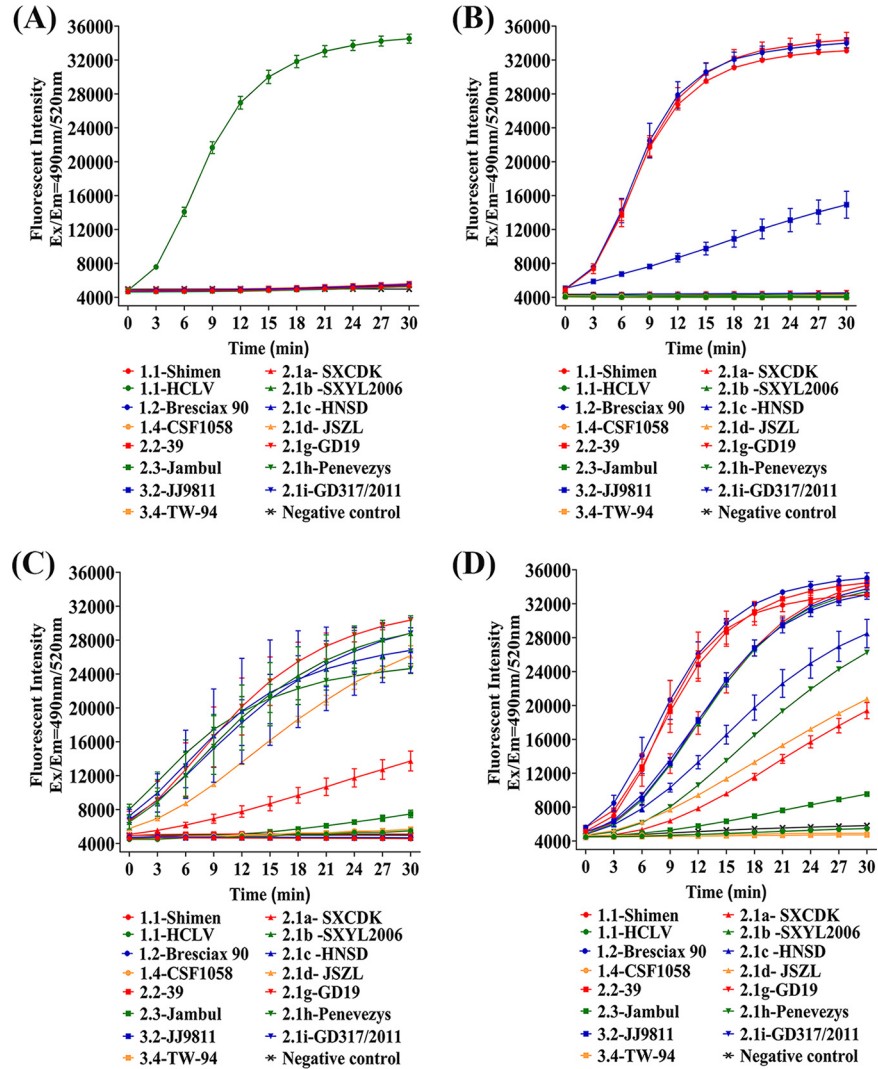

**FIG 2** Estimation of the ability of CRISPR/Cas13a to recognize different classical swine fever virus (CSFV) genotypes with screened crRNAs. (A) Estimation of the ability of CRISPR/Cas13a-crHCLV5 to recognize different CSFV genotypes. Fluorescent kinetics of CRISPR/Cas13a-crHCLV5 to recognize different CSFV genotypes are indicated as figure annotations. The negative control used diethyl pyrocarbonate (DEPC) water to replace template RNA, with other conditions of collateral cleavage assay the same as for experimental groups. (B) Estimation of the ability of CRISPR/Cas13a-crShimen7 to recognize different CSFV genotypes. Fluorescent kinetics of CRISPR/Cas13a-crShimen7 to recognize different CSFV genotypes are indicated as figure annotations. The negative control used DEPC water to replace template RNA, with other conditions of collateral cleavage assay the same as for experimental groups. (C) Estimation of the ability of CRISPR/Cas13a-cr2.17 to recognize different CSFV genotypes. Fluorescent kinetics of CRISPR/Cas13a-cr2.17 to recognize different CSFV genotypes are indicated as figure annotations. The negative control used DEPC water to replace template RNA, with other conditions of collateral cleavage assay the same as for experimental groups. (D) Estimation of the ability of CRISPR/Cas13a-crShimen7/cr2.17 to recognize different CSFV genotypes. Fluorescent kinetics of CRISPR/Cas13a-crShimen7/cr2.17 to recognize different CSFV genotypes are indicated as figure annotations. The negative control used DEPC water to replace template RNA, with other conditions of collateral cleavage assay the same as for experimental groups. Ex, excitation; Em, emission.

sequences of all virulent genotypes were further analyzed. Alignment results showed that there were three main mutations in subgenotype 2.1, including G to A at position 5, U to C at position 16, and U to A at position 19. Based on these mutations, cr2.17 was designed to detect CSFV all subsubgenotypes of 2.1. Collateral cleavage assay showed that even though the efficiencies of cr2.17 to recognize different subsubgenotypes of 2.1 were varied, which depends on the number and position of mutations, cr2.17 can differentiate all subsubgenotypes of 2.1 (including 2.1a, 2.1b, 2.1c, 2.1d, 2.1g, 2.1i, and 2.1h) from the vaccine strain (Fig. 2C).

For the purpose of detecting both the traditional virulent Shimen strain and the dominant pandemic virulent strains of subgenotype 2.1, crShimen7 and cr2.17 were both introduced in collateral cleavage assay. The results showed that except for subgenotypes 2.4 and 3.4, CRISPR/Cas13a-crShimen/cr2.17 recognized most virulent subgenotypes, including the traditional Shimen strain and all dominant pandemic virulent subsubgenotypes of 2.1 that have been reported to exist in China. In addition, CRISPR/Cas13a-crShimen7/cr2.17 showed no cross-reactivity to the HCLV strain, exhibiting the ability to differentiate virulent strains from the vaccine strain (Fig. 2D).

**Sensitivity of RT-RAA-CRISPR/Cas13a.** A recent study compared two recombinase-based techniques, recombinase polymerase amplification (RPA) and RAA, in parallel for rapid detection for African swine fever virus (ASFV), demonstrating that the detection limit for RPA was 93.4 copies/reaction, while the detection limit for RPA was 53.6 copies/reaction (19). Thus, RAA was introduced to improve detecting sensitivity for this study.

The detection limit of CRISPR/Cas13a was estimated with synthetic Shimen and HCLV 3′UTR RNA by collateral cleavage assay. The results showed that CRISPR/Cas13a alone did not yield a detectable signal at input concentrations below $3.0 \times 10^{10}$ copies/$\mu$L (Fig. 3A and B). To enhance detecting sensitivity of CRISPR/Cas13a, RT-RAA was introduced to preamplify tested samples before collateral cleavage assay (Fig. S5; Table S5). The results showed that the sensitivity of RT-RAA-CRISPR/Cas13a was greatly improved compared to CRISPR/Cas13a alone, with a detection limit of $3.0 \times 10^2$ copies/$\mu$L (Fig. 3C and D).

**Specificity of RT-RAA-CRISPR/Cas13a.** Before estimating the specificity of RT-RAA-CRISPR/Cas13a, cDNAs common porcine viruses or members of pestiviruses were first tested by nPCR, except for ASFV, which used certified reference genomic nucleic acid material as the template and was tested by OIE-recommended PCR. The results showed that all of the tested viruses can be efficiently amplified (Fig. S4A), confirming the reliability of all samples used in specificity estimation.

Collateral cleavage assay showed that when extracted genomic nucleic acids of viral cell cultures were used as the templates, RT-RAA-CRISPR/Cas13a was also able to differentiate between Shimen and HCLV. In addition, RT-RAA-CRISPR/Cas13a showed no cross-reactivity to other viruses, exhibiting good detecting specificity (Fig. 4A and B).

**Nucleic acid-extraction-free detection of RT-RAA-CRISPR/Cas13a.** In order to simplify the procedure of RT-RAA-CRISPR/Cas13a detection, the HUDSON technique was introduced for robust treatment of tested samples without the need to extract nucleic acids. To estimate the compatibility of HUDSON and RT-RAA-CRISPR/Cas13a, cell cultures of Shimen and HCLV strains were treated with HUDSON and directly used for collateral cleavage assay. Only slightly decreases of fluorescent signals were observed when HUDSON-treated samples were tested compared with cognate genomic nucleic acids, maintaining a comparable detecting sensitivity (Fig. 5A and B).

**Comparison of HUDSON-RT-RAA-CRISPR/Cas13a, antigen ELISA, and nPCR.** Shimen and HCLV cell cultures and 10-fold serial dilutions from $10^{-1}$ to $10^{-4}$ were tested by HUDSON-RT-RAA-CRISPR/Cas13a, nPCR and antigen enzyme-linked immunosorbent assay (ELISA) in parallel. For both HUDSON-RT-RAA-CRISPR/Cas13a and nPCR, positive results were obtained at a maximum dilution level of $10^{-1}$ for HCLV and $10^{-2}$ for Shimen (Fig. 6A and B), showing an equal sensitivity to nPCR (Fig. 6C). The detection limits for cell cultures of Shimen and HCLV were $3.5 \times 10^2$ and $1.8 \times 10^2$ copies/$\mu$L, respectively, which were coincident with the detection limits for testing synthetic RNA templates. For antigen ELISA, a positive result was obtained only when testing Shimen cell culture without dilution, while suspected positive results were obtained when testing Shimen cell culture with $10^{-1}$ dilution and HCLV cell culture without dilution, indicating that antigen ELISA was at least 100 times less sensitive than HUDSON-RT-CRISPR/Cas13a and nPCR (Fig. 6D).

Before comparison of these three methods for testing tissue samples, cDNA of 50 spleen samples was first tested by PCR with individual specific primers of bovine viral diarrhea virus (BVDV), atypical porcine pestivirus (APPV), porcine reproductive and respiratory syndrome

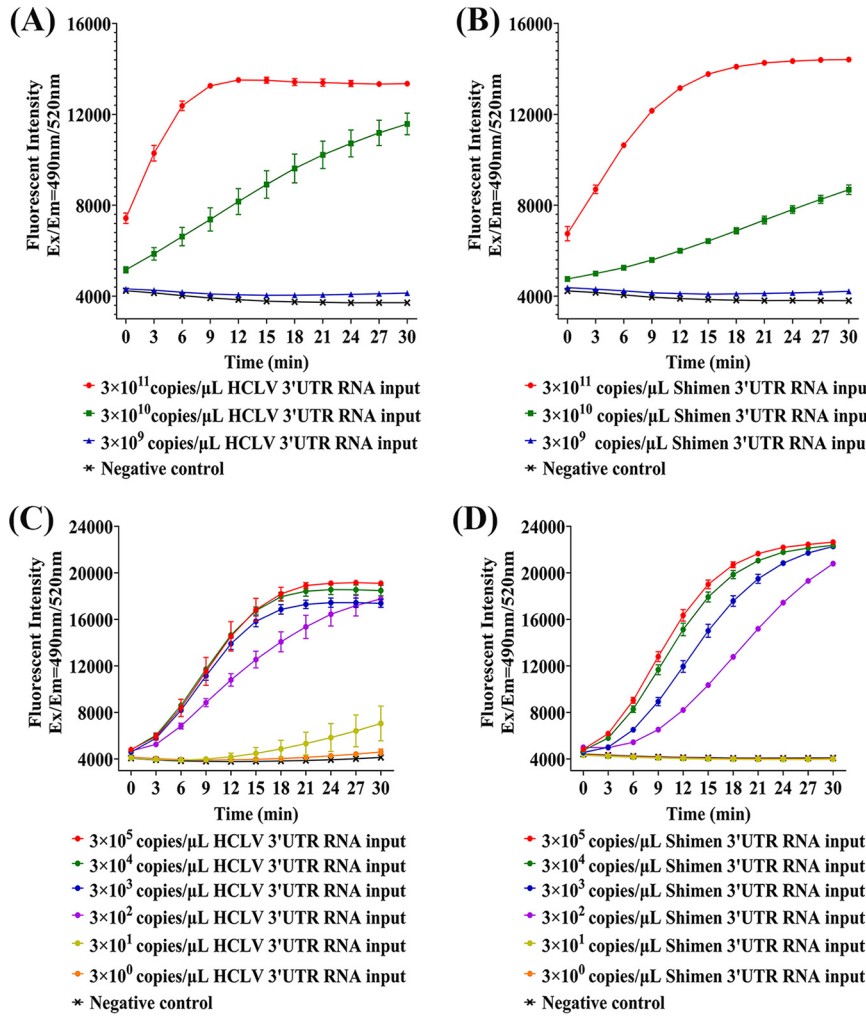

**FIG 3** Detecting sensitivity of CRISPR/Cas13a for synthetic CSFV 3'-untranslated region (3′UTR) RNA. (A) Detecting sensitivity of CRISPR/Cas13a-crHCLV5 for synthetic hog cholera lapinized virus (HCLV) 3′UTR RNA. Fluorescent kinetics of CRISPR/Cas13a-crHCLV5 in detecting HCLV 10-fold dilutions of synthetic 3′UTR RNAs are indicated as figure annotations. The negative control used DEPC water to replace template RNA, with other conditions of collateral cleavage assay the same as for experimental groups. (B) Detecting sensitivity of CRISPR/Cas13a-crShimen7/cr2.17 for synthetic Shimen 3′UTR RNA. Fluorescent kinetics of CRISPR/Cas13a-crShimen7/cr2.17 in detecting Shimen 10-fold dilutions of synthetic 3′UTR RNAs are indicated as figure annotations. The negative control used DEPC water to replace template RNA, with other conditions of collateral cleavage assay the same as for experimental groups. (C) Detecting sensitivity of RT-RAA-CRISPR/Cas13a-crHCLV5 for synthetic HCLV 3′UTR RNA. Fluorescent kinetics of RT-RAA-CRISPR/Cas13a-crHCLV5 in detecting HCLV 10-fold dilutions of synthetic 3′UTR RNAs are indicated as figure annotations. The negative control used DEPC water to replace template RNA, with other conditions of RT-RAA and collateral cleavage assay the same as for experimental groups. (D) Detecting sensitivity of RT-RAA-CRISPR/Cas13a-crShimen7/cr2.17 for synthetic Shimen 3′UTR RNA. Fluorescent kinetics of RT-RAA-CRISPR/Cas13a-crShimen7/cr2.17 in detecting Shimen 10-fold dilutions of synthetic 3′UTR RNAs are indicated as figure annotations. The negative control used DEPC water to replace template RNA, with other conditions of RT-RAA and collateral cleavage assay the same as for experimental groups.

virus (PRRSV), porcine epidemic diarrhea virus (PEDV), African swine fever virus (ASFV), pseudorabies virus (PRV), and porcine circovirus 2 (PCV2) to investigate the background of each sample (Fig. S4). The results showed that the infection situations were complex among these samples, some of which exhibited coinfection with more than one virus; 15 samples showed CSFV positive, including samples 22, 35 to 38, and 41 to 50. Unfortunately, nPCR for BVDV used in this study cannot differentiate BVDV from CSFV. Considering that primers for CSFV can only amplify CSFV while primers for BVDV can amplify both BVDV and CSFV, no BVDV positivity was found among the tested samples. Similarly, no APPV-infected sample was

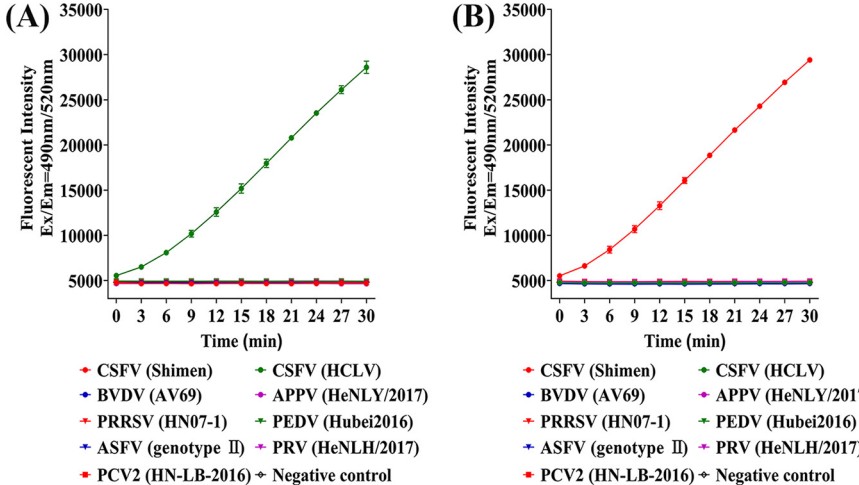

**FIG 4** Specificity of RT-RAA-CRISPR/Cas13a. (A) Specificity of RT-RAA-CRISPR/Cas13a-crHCLV5. Fluorescent kinetics of RT-RAA-CRISPR/Cas13a-crHCLV5 in detecting extracted genomic nucleic acids of different porcine virus cell cultures are indicated as figure annotations. The negative control used DEPC water to replace extracted genomic nucleic acids, with other conditions of RT-RAA and collateral cleavage assay the same as for experimental groups. (B) Specificity of RT-RAA-CRISPR/Cas13a-crShimen7/cr2.17. Fluorescent kinetics of RT-RAA-CRISPR/Cas13a-crShimen7/cr2.17 in detecting extracted genomic nucleic acids of different porcine virus cell cultures are indicated as figure annotations. The negative control used DEPC water to replace extracted genomic nucleic acids, with other conditions of RT-RAA and collateral cleavage assay the same as for experimental groups.

found. One sample (sample 27) was found to be PEDV positive, and 23 samples were PRRSV positive, including samples 02 to 06, 13, 16 to 20, 24 to 26, 28 to 34, 39, and 40. Because all of these samples were collected between 2016 and 2018, a period before the first ASF outbreak in China in 2019, no ASFV-positive sample was found. Three samples were PRV positive, including samples 07, 16, and 23. Twenty-seven samples were PCV2 positive, including 01, 03 to 05, 08 to 16, 18 to 21, 23 to 25, 31 to 34, 39, and 40.

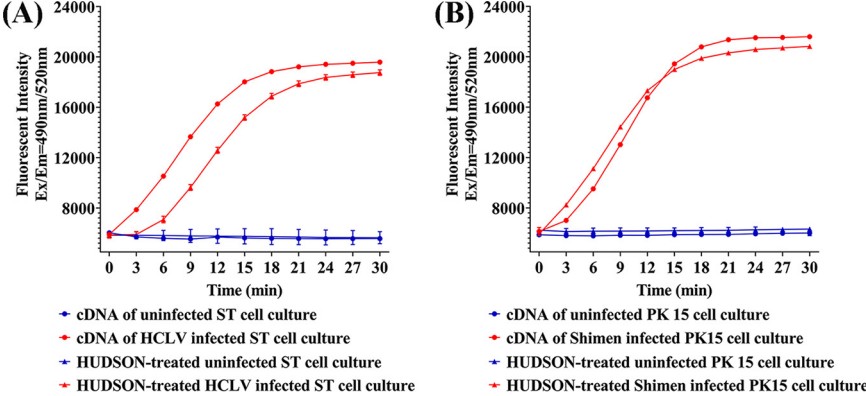

**FIG 5** Estimation of HUDSON-RT-RAA-CRISPR/Cas13a for direct CSFV cell culture detection. (A) Estimation of RT-RAA-CRISPR/Cas13a-crHCLV5 for direct HCLV ST cell culture detection. Comparison of RT-RAA-CRISPR/Cas13a-crHCLV5 in detecting cDNA and HUDSON-treated HCLV ST cell culture were assayed to estimate the influence of HUDSON treatment on positive fluorescent signal. Comparisons of RT-RAA-CRISPR/Cas13a-crHCLV5 in detecting cDNA and HUDSON-treated uninfected ST cell culture were assayed to estimate the influence of HUDSON treatment on background fluorescent signal. Fluorescent kinetics in detecting each kind of samples are indicated as figure annotations. (B) Estimation of RT-RAA-CRISPR/Cas13a-crShimen7/cr2.17 for direct Shimen PK15 cell culture detection. Comparison of RT-RAA-CRISPR/Cas13a-crShimen7/cr2.17 in detecting cDNA and HUDSON-treated Shimen PK15 cell culture were assayed to estimate the influence of HUDSON treatment on positive fluorescent signal. Comparisons of RT-RAA-CRISPR/Cas13a-crShimen7/cr2.17 in detecting cDNA and HUDSON-treated uninfected PK15 cell culture were assayed to estimate the influence of HUDSON treatment on background fluorescent signal. Fluorescent kinetics in detecting each kind of samples are indicated as figure annotations. ASFV, African swine fever virus; BVDV, bovine viral diarrhea virus; PCV 2, porcine circovirus 2; PEDV, porcine epidemic diarrhea virus; PRRSV, porcine reproductive and respiratory syndrome virus; PRV, pseudorabies virus.

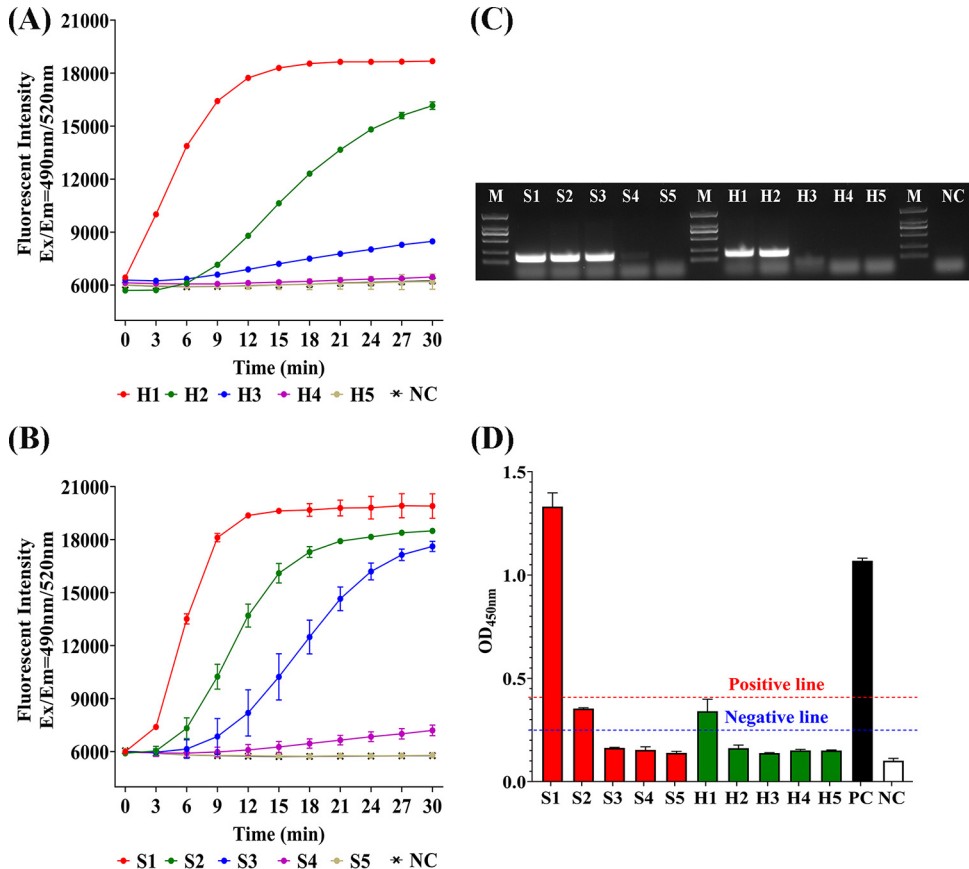

**FIG 6** Comparison of HUDSON-RT-RAA-CRISPR/Cas13a, nested PCR (nPCR) and antigen ELISA in detecting 10-fold dilutions of CSFV cell cultures. (A) HUDSON-RT-RAA-CRISPR/Cas13a-crHCLV5 detection for 10-fold dilutions of HCLV swine testis (ST) cell culture. Fluorescent kinetics in detecting each dilution are indicated as follows. H1, undiluted HCLV ST cell culture ($1.8 \times 10^4$ copies/$\mu$L); H2, $10^1$-fold diluted HCLV ST cell culture ($1.8 \times 10^3$ copies/$\mu$L); H3, $10^2$-fold diluted HCLV ST cell culture ($1.8 \times 10^2$ copies/$\mu$L); H4, $10^3$-fold diluted HCLV ST cell culture ($1.8 \times 10^1$ copies/$\mu$L); H5, $10^4$-fold diluted HCLV ST cell culture ($1.8 \times 10^0$ copies/$\mu$L); NC, negative control (uninfected ST cell culture). (B) HUDSON-RT-RAA-CRISPR/Cas13a-crShimen7/cr2.17 detection for 10-fold dilutions of Shimen porcine kidney 15 (PK15) cell culture. Fluorescent kinetics in detecting each dilution are indicated as follows. H1, Undiluted Shimen PK15 cell culture ($3.5 \times 10^5$ copies/$\mu$L); H2, $10^1$-fold diluted Shimen PK15 cell culture ($3.5 \times 10^4$ copies/$\mu$L); H3, $10^2$-fold diluted Shimen PK15 cell culture ($3.5 \times 10^3$ copies/$\mu$L); H4, $10^3$-fold diluted Shimen PK15 cell culture ($3.5 \times 10^2$ copies/$\mu$L); H5, $10^4$-fold diluted Shimen PK15 cell culture ($3.5 \times 10^1$ copies/$\mu$L); NC, negative control (uninfected PK15 culture). (C) Agarose gel electrophoresis of CSFV nPCR detection for 10-fold dilutions of HCLV ST cell culture or Shimen PK15 cell culture. M, Tans2K DNA marker (TransGege, China); H1 to H5, 10-fold dilutions of HCLV ST cell culture, as indicated in panel A. Columns S1 to S5, 10-fold dilutions of HCLV ST cell culture, as indicated in panel B; NC, negative control (cDNA of uninfected ST cell culture). (D) CSFV antigen ELISA detection for 10-fold dilutions of HCLV ST cell culture or Shimen PK15 cell culture. H1 to H5, 10-fold dilutions of HCLV ST cell culture, as indicated in panel A; columns S1 to S5, 10-fold dilutions of HCLV ST cell culture, as indicated in panel B; columns PC and NC, positive control and negative control provided by ELISA kit to determine positive line, negative line, and suspected section.

A total of 15 samples tested positive, and 35 samples were negative by HUDSON-RT-RAA-CRISPR/Cas13a, which showed 100% coincident rate with nPCR. However, due to the lower detecting sensitivity of antigen ELISA, among these 15 CSFV-positive samples, only 6 samples tested positive, while 4 samples were suspected positive, and another 5 samples tested negative by antigen ELISA (Table S6; Fig. S4), showing 82.0% coincident rate (Table 1).

## DISCUSSION

Accurate diagnosis of important infectious diseases has been paid increasing attention in recent years, especially since the outbreaks of ASF and COVID-19 that have brought about enormous economic losses worldwide (20, 21). With the development

**TABLE 1** Comparison of HUDSON-RT-RAA-CRISPR/Cas13a, nested PCR and antigen ELISA in testing 50 spleen samples[a]

| Method | Positive (+) | Suspected (±) | Negative (−) |
|---|---|---|---|
| HUDSON-RT-RAA-CRISPR/Cas13a | 15 | 0 | 35 |
| Nested PCR | 15 | 0 | 35 |
| Antigen ELISA | 6 | 4 | 40 |

[a]ELISA, enzyme-linked immunosorbent assay; HUDSON, heating unextracted diagnostic samples to obliterate nucleases; RT-RAA, reverse transcription recombinase-aided amplification.

of science and technology, continuously emerging techniques have been developed to meet specific diagnostic demands of various infectious diseases that traditional methods could not achieve. Due to the impressive ability to precisely recognize nucleic acids, CRISPR/Cas9 has won the Nobel prize as a powerful gene-editing tool for rewriting the code of life (22). Similarly, as another member of the Cas family, Cas13a is also able to precisely recognize RNA with single-base specificity. With the unique collateral RNase activity, Cas13a is first effector that transforms CRISPR technology from gene scissor into gene sensor (17).

In this study, we established CRISPR/Cas13a platform for CSFV DIVA application, which is one of the greatest obstacles for CSF eradication. Even though CRISPR/Cas13a alone is not sensitive enough, RT-RAA significantly increased the detecting sensitivity, with a level equal to that of nPCR. Combining the high specificity of CRISPR with the high sensitivity of RAA is a promising platform for differential diagnosis of infectious diseases. Due to the simplicity of primer requirement and base mismatch tolerance of RAA (23), it is easy to design universal primers to detect a broad spectrum of all genotypes of a specific virus like CSFV, followed by differential detection by CRISPR/Cas13a.

In addition, both RAA and CRISPR collateral cleavage assay were applied at a constant 37°C, showing the potential to be developed as point-of-care testing tools. By simply heating samples with EDTA and Tris(2-carboxyethyl)phosphine hydrochloride (TCEP), HUDSON treatment can robustly release nucleic acids and inactivate ribonucleases for downstream molecular detection. It can be further combined with commercial lateral flow strips and biotin-6-carboxyfluorescein (FAM)-labeled polyU reporter to replace fluorescent readout with visual readout, achieving complete instrument-free detection from sample processing to diagnostic results (24).

In addition to Cas13a, Cas12a is another CRISPR effector with collateral DNase activity that has been widely developed as a gene-sensing tool in recent years (25). Cas12a is able to directly target DNA molecules, it shows advantages beyond Cas13a as a more stable, simpler, and less expensive tool. However, Cas13a has fewer sequential restrictions than Cas12a, with a protospacer flanking site requirement (not G) rather than protospacer adjacent motif requirement (TTTV). In this study, a 50-nucleotide (nt)-long region from the CSFV 3′UTR area contains nine candidate Cas13a crRNAs for Shimen strain and five crRNAs for HCLV strain. While no suitable Cas12a crRNA was found in the same region, Cas13a exhibited little dependency to target sequence than Cas12a. Different characters of these two effectors make Cas13a more suitable for differential diagnosis with the two-step procedure established in this study, while Cas12a is more suitable for one-pot assay (26–28) or amplification-free detection (29–32) that has been presented in other works. With an increasing number of Cas effectors with various characters to be explored and developed as gene-sensing tools, studies about the CRISPR method of diagnosing infectious diseases is a promising direction for the future.

**Conclusion.** In this study, we established CRISPR/Cas13a platform for differentiation between CSFV virulent and vaccine strains with high sensitivity and good specificity. We also introduced HUDSON treatment to replace nucleic acids extraction step, facilitating the detection procedure and reducing detection time. The established platform can be performed at a constant 37°C, showing the potential to be developed as point-of-care testing. Future work will focus on development of convenient readout methods, such as lateral flow assay or portable fluorescent devices.

## MATERIALS AND METHODS

**Plasmids, viruses, and clinical samples.** *Leptotrichia wadei* Cas13a (LwaCas13a) bacterial expression plasmid Twinstrep-SUMO-LwaCas13a was purchased from Addgene (https://www.addgene.org/90097/) and transformed into Rosetta (DE3) competent cells. Positive clones were verified by sequencing and preserved as bacterial glycerol stock at −80°C.

Complete CSFV 3′UTR of different genotypes (Table S1) with an upstream T7 promoter sequence (5′-TAATACGACTCACTATAGGGG-3′) were synthesized by Sangon Biotech and cloned into PUC57 plasmid. Other DNA oligonucleotides, including primers and crRNA transcription templates, were also synthesized by Sangon Biotech (China).

All viral cell cultures and tissue samples were provided by the Key Laboratory of Animal Immunology of Henan Academy of Agricultural Sciences, including the cell culture of CSFV, viral diarrhea virus (BVDV), porcine reproductive and respiratory syndrome virus (PRRSV), porcine epidemic diarrhea virus (PEDV), pseudorabies virus (PRV), and porcine circovirus 2 (PCV2), as well as an atypical porcine pestivirus (APPV) positive cDNA sample (Table S2). Fifty spleen tissues were collected during 2016 to 2018 from pigs with one or more suspected clinical signs, including pyrexia, huddling, weakness, conjunctivitis, and diarrhea. In addition, national certified reference material of African swine fever virus (ASFV) standard genomic DNA (genotype II) was purchased from the China Animal Health and Epidemiology Center.

**Soluble expression and purification of LwaCas13a.** LwaCas13a expression bacterial glycerol stock was inoculated LB medium for 2.5 h at 37°C and then induced with 0.5 mmol/L isopropyl-$\beta$-D-thiogalactopyranoside (IPTG) for 15 h at 16°C. Cell pellets were harvested and resuspended in lysis buffer (20 mmol/L Tris-HCl, pH 8.0, 500 mmol/L NaCl, 1 mmol/L dithiothreitol [DTT]). After sonication, the supernatant was collected for nickel affinity chromatography. The 5-mL nickel Sepharose column (GE Healthcare, USA) was washed with lysis buffer containing 20 mmol/L imidazole to remove nontarget proteins, followed by the elution of LwaCas13a with lysis buffer containing 100 mmol/L imidazole.

The elution fractions were then incubated with SUMO protease (Solarbio, China) for 8 h at 4°C to cleave off the His$_6$-Twinstrep-SUMO tag. After SUMO protease digestion, the proteins were diluted with SP elution buffer (20 mmol/L Tris-HCl, pH 8.0, 1 mmol/L DTT, 5% glycerol) in a volume ratio of 1:1. Diluted proteins were then loaded onto a 1-mL HiTrap SP cation exchange column (GE Healthcare, USA) and eluted over a NaCl gradient from 250 mmol/L to 1 mol/L. Fractions containing LwaCas13a were analyzed by SDS-PAGE, pooled, and stored at −80°C for further study.

**Preparation of CSFV 3′UTR RNA and crRNA.** CSFV genomic cDNA alignment of Shimen strain and HCLV strain showed a 12-nt insertion (5′-CTTTTTTCTTTT-3′) in HCLV 3′UTR region compared with Shimen 3′UTR region, which was coincident with previous study (33). Candidate crRNAs were designed by individually submitting partial 3′UTR regions of Shimen strain (5′-ACCCUAUUGUAGAUAACACUAAU UUUUAUUUAUUUA-3′) and HCLV strain (5′-UACACUACUUUUCUUUUCUUUUUUUAUUUUAUU-3′) to CRISPR-RT (Table S3) (34). Candidate crRNAs were *in vitro* transcribed by the T7 RiboMAX Express large-scale RNA production system (Promega, USA) and purified by NucAway spin columns (Thermo Fisher, USA) according to the manufacturer's instructions. Double-stranded DNA templates for transcription were prepared by annealing T7 promoter appending forward DNA oligonucleotide with its complementary reverse DNA oligonucleotide in annealing buffer (Beyotime Biotech, China). Each oligonucleotide pair was annealed by preheating the annealing cocktail (nuclease-free water 40 $\mu$L, 5$\times$ annealing buffer 20 $\mu$L, 50 $\mu$mol/L forward and reverse oligonucleotide 20 $\mu$L each) at 95°C for 2 min and then gradient decreasing 1°C every 90 s from 94 to 25°C.

PUC57 plasmids that contained CSFV 3′UTR DNA sequences of different genotypes were used as the templates for PCR amplification with forward primer (5′-TAATACGACTCACTATAGGGG-3′) and reverse primer (5′-TTAGGAAATTACCTTAGTCCAAC-3′) in a 50-$\mu$L cocktail containing 2$\times$ PrimeSTAR Max Premix (TaKaRa, China) 25 $\mu$L, 10 $\mu$mol/L forward and reverse primer 2.5 $\mu$L each, double-distilled water 18 $\mu$L and PUC57 plasmid 2 $\mu$L. The PCR steps were as follows: initial denaturation at 98°C for 1 min; 30 cycles at 98°C for 10 s, 45°C for 5 s, and 72°C for 30 s; and a final extension at 72°C for 5 min. PCR products were purified by Cycle Pure kit (Omega Bio-tek, China) and directly used as *in vitro* transcription templates. CSFV 3′UTR RNAs were then transcribed *in vitro* by T7 RiboMAX Express large-scale RNA production system (Promega, USA) and purified by MEGAclear transcription clean-up kit (Thermo Fisher, USA) according to the manufacturer's instructions.

**cDNA preparation of viral cell cultures and tissue samples.** Spleen tissue homogenates were prepared with a grinder in a ratio of 1:5 (tissue weight/phosphate-buffered saline [PBS] volume, mg/$\mu$L). Then, 500-$\mu$L viral cell cultures or spleen tissue homogenates were freeze-thawed three times and centrifuged at 12,000 $\times$ $g$ for 20 min. All procedures were performed at 4°C.

Total viral nucleic acids were extracted with MiniBEST Viral RNA/DNA extraction kit version 5.0 (TaKaRa, China). Briefly, 300 $\mu$L supernatant was used for extraction procedures according to the manufacturer's instruction, followed by elution with 30 $\mu$L nuclease-free water, and immediately used for cDNA preparation or stored at −80°C for further experiments.

cDNAs were prepared with PrimeScript RT Master Mix (TaKaRa, China) according to the manufacturer's instructions. A 10-$\mu$L reverse transcription cocktail contained 5$\times$ PrimeScript RT Master Mix (2 $\mu$L), 1 $\mu$L total viral nucleic acids (less than 500 ng), and 7 $\mu$L nuclease-free water. Once reverse transcription was done, cDNAs were stored at −80°C for further experiments.

**LwaCas13a collateral cleavage assay.** A single 50-$\mu$L cocktail contained 1$\times$ reaction buffer (20 mmol/L HEPES, pH 6.8, 60 mmol/L NaCl, 6 mmol/L MgCl$_2$), 50 nmol/liter LwaCas13a, 25 nmol/liter crRNA, 1.6 unit/$\mu$L murine RNase inhibitor (New England Biolabs, USA), 125 nmol/liter quenched fluorescent RNA reporter (Thermo Fisher, USA), and various amounts of input nucleic acid target (5 $\mu$L). If input nucleic acid target was amplified DNA, the above reaction was modified to include 0.5 $\mu$L T7

transcription enzyme mix and 2.5 $\mu$L T7 transcription buffer (Promega, USA), as well as 2 $\mu$L amplified DNA. LwaCas13a collateral cleavage assay was proceeded for 30 min at 37°C on POLARstar Omega multifunction reader (BIO-GENE Biotech, China) with fluorescent kinetics (excitation/emission [Ex/Em] = 490 nm/520 nm) measured every 3 min.

**RT-RAA and PCR.** RT-RAA was performed with basic RT-RAA kit (ZC Bioscience, China) according to the manufacturer's instructions. A 50-$\mu$L RT-RAA cocktail included 41.5 $\mu$L rehydration buffer (buffer A), 1 $\mu$L each of 20 $\mu$mol/L T7 promoter appending forward primer (5'-TAATACGACTCACTATAGGGGGGG AACCCGCCAGTAGGACCCTATTGTAGATA-3') and reverse primer (5'-GTGGTAACTTGAGGTAGTTTGTACC AGTTCTT-3'), 4 $\mu$L RNA template, and 2.5 $\mu$L 280 mmol/L MgOAc (buffer B). RT-RAA cocktail was performed at 37°C in a water bath for 30 min and immediately used as amplified DNA for LwaCas13a collateral cleavage assay.

CSFV nested PCR (nPCR) was performed according to OIE recommend and previous study with some modifications (35). Primary PCR was performed with outer primer pairs (outer-forward-primer 5'-CAACTGGCTVGTYAAYGC-3' and outer-reverse-primer 5'-AATGAGTGTAGTGTGGTAAC-3', V = A or G; Y = C or T) in a 25-$\mu$L cocktail, which contained 12.5 $\mu$L 2× rTaq mix (TaKaRa, China), 1 $\mu$L 10 $\mu$mol/L outer forward or reverse primer each, 8 $\mu$L double-distilled water, and 2.5 $\mu$L 10-fold diluted cDNA. The first round PCR steps were as follows: initial denaturation at 95°C for 1 min; 25 cycles at 94°C for 30 s, 54°C for 30 s, and 72°C for 30 s; and a final extension at 72°C for 10 min. Amplified DNA of first round PCR was then used as the templates for second-round PCR, with inner primer pairs (inner-forward-primer 5'-ATGATGATGVCSCTKATA-3' and inner-reverse-primer 5'-GTGTGGTAACWTGAGGTAG-3', V = A or G; Y = C or T, V = A or G; S = C or G; K = T or G; W = A or T) in a 25-$\mu$L cocktail, which contained 12.5 $\mu$L of 2× rTaq mix, 1.5 $\mu$L of 10 $\mu$mol/L inner forward or reverse primer, 9 $\mu$L of double-distilled water, and 0.5 $\mu$L of first round PCR product. The second-round PCR steps were as follows: initial denaturation at 95°C for 3 min; 35 cycles at 94°C for 30 s, 56°C for 30 s, and 72°C for 20 s; and a final extension at 72°C for 10 min. In addition, the primers and conditions of nPCRs for BVDV, APPV, PEDV, PRRSV, PRV, and PCV2, as well as ASFV OIE-recommended PCR, are listed in Table S4.

**CSFV antigen ELISA.** CSFV antigen ELISA was performed with a commercial classical swine fever virus antigen test kit/serum plus (IDEXX, China). Supernatants of viral cell cultures or tissue homogenates were mixed with detection solution in a volume ratio of 1:1 and incubated at 37°C for 2 h, followed by adding conjugate to each well and incubated at room temperature for another 30 min. Color development was performed with TMB substrate at room temperature for 10 min and stopped by stop solution. The results were calculated according to the manufacturer's instructions. In brief, optical density at 450 mm ($OD_{450}$) values of negative-control (N), positive-control (P), and tested samples (S) were measured with POLARstar Omega multifunction reader (BIO-GENE Biotech, China). For a reliable experiment, the N value should be less than 0.250, while the value of (P – N) should more than 0.150. The results were then calculated according to the following rules: a positive result was determined if the value of (S-N) was more than 0.300; a suspected result was determined if the value of (S – N) was between 0.100 and 0.300; and a negative result was determined if the value of (S – N) was less than 0.100.

**HUDSON treatment.** HUDSON was performed for robust treatment of tested samples according to previous study (18). In brief, supernatants of viral cell cultures or tissue homogenates were mixed with EDTA and TCEP at final concentrations of 1 and 100 mmol/L, respectively, followed by boiling for 5 min to lyse viral particles and inactivate nucleases. The mixture then was directly used for RT-RAA and LwaCas13a collateral cleavage assay.

## SUPPLEMENTAL MATERIAL

Supplemental material is available online only.
**SUPPLEMENTAL FILE 1**, PDF file, 0.6 MB.

## ACKNOWLEDGMENTS

This work was supported by grant 2016YFD0500701 from the National Key Research and Development Program of China, grant CARS-35 from the Earmarked Fund for Modern Agro-Industry Technology Research System of China, grant S2012-06-02 from the Special Fund for Henan Agriculture Research System, grant 222102110453 from Science and Technology Development Project of Henan Province, and grants 2021TD02 and 2022TD02 from Science and Technology Innovation Team of Henan Academy of Agricultural Sciences.

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
