## [Reviewer comments · Microbiology Spectrum]

Microbiology Spectrum

Differentiation of Classical Swine Fever Virus Virulent and Vaccine Strains by CRISPR/Cas13a

Yuhang Zhang, Qimei Li, Ruining Wang, Li Wang, Xun Wang, Jun Luo, Guangxu Xing, Guanmin Zheng, Bo Wan, Junqing Guo, and Gaiping Zhang

Corresponding Author(s): Gaiping Zhang, Key Laboratory of Animal Immunology, Henan Academy of Agricultural Sciences

Review Timeline:

Submission Date:	March 11, 2022
Editorial Decision:	April 2, 2022
Revision Received:	June 14, 2022
Editorial Decision:	June 18, 2022
Revision Received:	July 8, 2022
Accepted:	July 10, 2022

Editor: Biao He

Reviewer(s): Disclosure of reviewer identity is with reference to reviewer comments included in decision letter(s). The following individuals involved in review of your submission have agreed to reveal their identity: Hua-Ji Qiu (Reviewer #1); Qin Zhao (Reviewer #3)

Transaction Report:

DOI: <https://doi.org/10.1128/spectrum.00891-22>

April 2, 2022

Dr. Gaiping Zhang
Key Laboratory of Animal Immunology, Henan Academy of Agricultural Sciences
No.116 Huayuan Road
Zhengzhou
China

Re: Spectrum00891-22 (Differentiation of Classical Swine Fever Virus Virulent and Vaccine Strains by CRISPR/Cas13a)

Dear Dr. Gaiping Zhang:

Thank you for submitting your manuscript to Microbiology Spectrum. Your manuscript has been reviewed by three experts with well expertise in the field. As you can see below, reviewers concerned the size of clinical sample and the ability to differentiate other pestiviruses, and the reaction conditions are expected to be optimized to improve the sensitivity of this method. Based on their comments, my decision at this time is 'Modification'.

Link Not Available

Sincerely,

Biao He

Journals Department
Reviewer comments:

Reviewer #1 (Comments for the Author):

Comments on Spectrum00891-22

Title: Differentiation of Classical Swine Fever Virus Virulent and Vaccine Strains by CRISPR/Cas13a

The present study by Yuhang Zhang et al. developed a CRISPR/Cas13a-based novel method for differential detection of wild-type classical swine fever virus and C-strain. The method was evaluated to be sensitive and specific. However, several concerns

need to be addressed.

1. More reference CSFV strains of different genotypes and more clinical samples of different types should be included to evaluate the established method.
2. Other pestiviruses, including atypical porcine pestivirus and border disease virus, need to be included in the specificity test.
3. The figures and tables should be improved in quality and presentation. Some figures and tables can be merged, removed, or presented as supplementary materials. All figures should be labeled with serial numbers for discrimination.
4. The manuscript should be revised by native English speakers.

Reviewer #2 (Comments for the Author):

In this study, the authors established a method for the differentiation of Classical swine fever virus virulent and vaccine strains by using HUDSON-RT-RAA-CRISPR/Cas13a, the sensitivity and specificity were also evaluated. However, the present assay established need be further evaluated. As a CRISPR detection method, the sensitivity in this assay just only reaches 300 copies/ μ L. At present, I have checked many publications on CRISPR detection methods, and the detection limit can be reached 10 copies or even 1 copy per μ L. Additionally, the comparative experiment with ordinary PCR method is done in this assay, and the sensitivity is the same each other, which causes that this assay has no practical significance in the field, and the sensitivity is not up to that of qPCR. Therefore, I suggest further optimization of the method. Other points as follow:

1. In Figure 1, the detection method is divided into 3 steps. It is whether the reagent is added by opening the lid according to the steps? If it is added in steps, it is concerned about the possible nucleic acid aerosol contamination after opening the lid between the second and third steps.
2. In Figure 2, the annotation of marker does not match the figure .
3. In Figure 3, the representation of the 3A and 3B is reversed.
4. In Figure 9, many non-specific bands can be seen, and I have doubts about the results of this PCR.
5. It is recommended to incorporate some figures and tables to enhance the readability of the manuscript.
6. Line 309, cNDA should be modified to cDNA ; Line 311 ,PCV 2 should remove the space.

Reviewer #3 (Comments for the Author):

In the present study, Zhang et al. reports a CRISPR/Cas13a based diagnostic method for differential detection of virulent and vaccine strains of classical swine fever virus (CSFV). Considering CSF eradication is undergoing in China and countries that use hog cholera lapinized virus (HCLV) derived live attenuated live vaccines, this method has important research significance for it provides a promising strategy for CSF diagnosis and eradication. However, there are still some major and minor concerns that need author to be improved as following:

Major concerns:

1. Line 33, " In combination with recombinase-aided amplification (RAA) ". As it is known, there are currently two recombinase based amplification techniques, including recombinase polymerase amplification (RPA) and recombinase-aided amplification (RAA), of which RPA is the most popular amplification technique that combined with CRISPR. What are the differences between RPA and RAA? Is there any reason that RAA is chose in the present manuscript?
2. Line 115, " 50 spleen tissues collected from pigs with suspected clinical signs". This clinical signs should be described more specifically. In other words, what signs that are likely to be caused by CSFV and may confused with other porcine virus infection.
3. Line 163, " Viral cell cultures or tissue honogenates (100 mg sample in 500 μ L PBS)". It is confused to understand how tissue honogenates were prepared, a more detailed description is needed to be clarified.
4. Line 219, figure 8 and figure 11, " results were calculated according to the manufacturer's instruction". There is a need to introduce how to decide the ELISA results. Similarly, how to determine the positive and negative lines in figure 8 and figure 11?

Minor concerns:

1. There are some abbreviations such as CSFV, HUDSON, etc., should be given the full name when they appeared for the first time.
2. Line 110, "synthesized by Sangon Biotech". Please indicate the country of the company. In addition, all regents and company in this manuscript should also indicate country.

Staff Comments:

Preparing Revision Guidelines

Please return the manuscript within 60 days; if you cannot complete the modification within this time period, please contact me. If you do not wish to modify the manuscript and prefer to submit it to another journal, please notify me of your decision immediately so that the manuscript may be formally withdrawn from consideration by Microbiology Spectrum.

Revision Notes

Dear Editor:

Thank you for your kind letter of “Spectrum00891-22 Decision Letter” on April 4th, 2022.

Generally, we appreciate the editor and reviewer’s insightful comments, which were helpful for improving the manuscript. Based on your comments and requests, we have made modification on the original manuscript.

Here are some issues need to communicate with editor:

1. The name of author’s department 1 was more specifically changed from “College of Veterinary Medicine, Henan Agricultural University, Zhengzhou, China” to “International Joint Research Center of National Animal Immunology, College of Veterinary Medicine, Henan Agricultural University, Zhengzhou, China”.
2. Bo Wan, the vice professor from Henan Agricultural University, has made great contribution during processing the revised manuscript, including providing necessary experimental materials and improving manuscript’s quality. Thus, we decided to listed Bo Wan as an additional co-author.
3. There is a mistake of second co-author. During the first vision of the manuscript, the name of second co-author is “Qimei Li” on the website of the journal, while the correct name is “Qingmei Li”. We failed to correct the name online, and we are hopeful to ask editor’s kind help to correct the name.
4. As listed in the manuscript, there are two corresponding authors in the manuscript. The first corresponding author is Gaiping Zhang, while the second one is Junqing Guo. However, we cannot choose Junqing Guo as corresponding author online. We’d like to know if Microbiology Spectrum only allows one corresponding author? If not, we are hopeful to ask editor’s kind help to choose Junqing Guo as second corresponding author.

Thanks again for insightful comments from editor and reviewers. We have modified the manuscript and made point-to-point response according to reviewers’ comments. In revision notes, the line numbers refer to the PDF vision of the revised manuscript.

Responses to reviewer #1 comments:

We have highlighted **the changes** in yellow according to reviewer #1 comments in the revised manuscript.

The present study by Yuhang Zhang et al. developed a CRISPR/Cas13a-based novel method for differential detection of wild-type classical swine fever virus and C-strain. The method was evaluated to be sensitive and specific. However, several concerns need to be addressed.

Many thanks. We appreciate the reviewer's carefulness and insightful comments. Some details have been clarified as follows.

1. More reference CSFV strains of different genotypes and more clinical samples of different types should be included to evaluate the established method.

Re: We are thankful to the reviewer's insightful comments. Currently, there are three genotypes and eleven subgenotypes based on partial sequences of 5'UTR, E2 and NS5B. However, limited information of 3'UTR is available on NCBI and published studies. In the present manuscript, we systematically estimated the ability of established CRISPR/Cas13a to recognize all CSFV genotypes whose sequential information can be found, including subgenotype 1.1, 1.2, 1.4, 2.2, 2.3, 3.2 and 3.4, as well as sub-subgenotype 2.1a, 2.1b, 2.1c, 2.1d, 2.1g, 2.1h and 2.1i. Though subgenotype 1.3, 3.1, 3.3 as well as sub-subgenotype 2.1e, 2.1f cannot be assayed currently, the predominant genotypes reported once occurred in China were tested, all of which can be recognized by established CRISPR/Cas13a in this study.

During development of CRISPR/Cas13a, we actually tested different organs of two infected pigs, including heart, liver, spleen, lung, kidney, brain and lymphatic node. Our results showed that spleen and lymphatic node from both pigs showed the highest fluorescent signals. In addition, in comparison experiment of clinical samples between CRISPR/Cas13a, nPCR and antigen ELISA, we further considered spleen was one major virus carrier for most porcine diseases, which means spleen was most likely to be infected with different viruses, thus we obtained fifty spleen samples to estimate the performance of established CRISPR/Cas13a assay.

2. Other pestiviruses, including atypical porcine pestivirus and border disease virus, need to be included in the specificity test.

Re: We are thankful to the reviewer's insightful comments. In the previous manuscript, only one pestivirus, bovine viral diarrhoea virus (BVDV), was assayed in the specificity test, which might

not convinable to declare the specificity of the established CRISPR/Cas13a method. Thus, we prepared cDNA from one porcine tissue infected by atypical porcine pestivirus (APPV), verified it by PCR and then tested it with CRISPR/Cas13a. Results showed the prepared cDNA was verified as APPV-positive by PCR, while CRISPR/Cas13a showed no cross-reactive with this sample.

As another member of pestiviruses, even though border disease virus (BDV) is less harmful to pigs compared with APPV and BVDV, it is also considered to be tested by CRISPR/Cas13a. We keep contacting with different colleges and organizations to search for a BDV positive sample once we received comments of reviewer #1. Unfortunately, we failed to obtain one at present. Cas13a is a CRISPR effector with high specificity for molecular sensing, only tolerating no more than two mismatches. We aligned our crRNAs with the genome of BDV published on NCBI (GenBank: NC_003679.1) by snapgene software. Results showed that no binding site was found throughout the genome, indicating CRISPR/Cas13a theoretically show no cross-reaction with BDV. Indeed, it is not convinable to use bioinformatic ways to assess the specificity. We are still devoting to seek BDV positive samples, hoping to provide experimental data in further revision if needed.

Added content in Line 60: *atypical porcine pestivirus (APPV).*

Added content in Line 137: *as well as an atypical porcine pestivirus (APPV) positive cDNA sample.*

Added content in Line 233 and Line 345: *APPV*

Added content in table 02: information of tested APPV cDNA sample

Added content in table 04: primer sequence and PCR condition for APPV.

Added content in figure S2: PCR detection for an APPV positive cDNA sample and 50 clinical samples with APPV specific primers.

Added content in line 352: *Similarly, no APPV infected sample was found as well.*

3. The figures and tables should be improved in quality and presentation. Some figures and tables can be merged, removed, or presented as supplementary materials. All figures should be labeled with serial numbers for discrimination.

Re: We are thankful to the reviewer's insightful comments. We have merged figure figure 9, figure 10 and figure 11 as one new **figure S2** and presented as supplementary materials. Alternatively, we reorganized results of comparison experiment of nPCR,

HUDSON-RT-RAA-CRISPR/Cas13a and antigen ELISA for testing 50 clinical samples and presented as **table 5**. In addition, all figures in the revised manuscript have been labeled with serial numbers.

4. The manuscript should be revised by native English speakers.

Re: We are thankful to the reviewer's insightful comments. We have asked for native English speakers and colleagues who once lived in English-speaking countries to revise throughout this manuscript. Detailed modifications are listed below:

Corrected mis-spellings:

Line 162: "*patial*" was changed to "*partial*".

Line 191: "*immeidately*" was changed to "*immediately*".

Line 202: "*apmlified*" was changed to "*amplified*".

Line 203: "*ezyme*" was changed to "*enzyme*".

Corrected grammatical mistakes:

Line 255-256: "*followed by boiling for 5 min to lysis viral particles and inactivate nucleases*" was changed to "*followed by boiling for 5 min to lyse viral particles and inactivate nucleases*".

Line 234: "*as well as ASFV OIE recommended PCR were lists in **Table 4***" was changed to "*as well as ASFV OIE recommended PCR were listed in **Table 4***".

Line 306-307: "*To enhance detection sensitivity of CRISPR/Cas13a*" was changed to "*To enhance detecting sensitivity of CRISPR/Cas13a*".

Line 338-339: "*which were coincident with the detection limits when testing synthetic RNA templates*" was changed to "*which were coincident with the detection limits for testing synthetic RNA templates*".

Improved writings:

Line 275-276: "*To estimate the ability of crHCLV5 and crShimen7 to distinguish between CSFV vaccine strain and virulent strains*" was changed to "*To estimate the ability of crHCLV5 and crShimen7 to differentiate between CSFV vaccine strain and virulent strains*".

Line 285-286: "*cr2.17 was designed to detect CSFV subgenotypes 2.1*" was changed to "*cr2.17 was designed to detect CSFV all sub-subgenotypes of 2.1*".

Line 286-287: "*Collateral cleavage assay showed that even though the efficiency of cr2.17 to recognize different subgenotype were varied*" was changed to "*Collateral cleavage assay showed*".

that even though the efficiency of cr2.17 to recognize different sub-subgenotypes of 2.1 were varied".

Line 290-291: *"For the purpose to detect both classical virulent Shimen strain and dominant pandemic virulent strains of subgenotype 2.1"* was changed to *"For the purpose to detect both traditional virulent Shimen strain and dominant pandemic virulent strains of subgenotype 2.1"*.

Line 278-280: *"including traditional Shimen strain and all dominant pandemic virulent 2.1 subgenotypes existed in China"* was changed to *"including traditional Shimen strain and all dominant pandemic virulent sub-subgenotypes of 2.1 that have been reported to exist in China"*.

Line 308-309: *"Results showed RT-RAA-CRISPR/Cas13a detection sensitivity was greatly improved"* was changed to *"Results showed the sensitivity of RT-RAA-CRISPR/Cas13a was greatly improved compared to CRISPR/Cas13a alone"*.

Line 313-316: *"Before estimating the detection specificity of RT-RAA-CRISPR/Cas13a, total genomic nucleic acids of common viral cell cultures were firstly detected by nPCR, except for ASFV that was used certified reference genomic nucleic acids as template and detected by OIE recommended PCR"* was changed to *"Before estimating the specificity of RT-RAA-CRISPR/Cas13a, cDNAs common porcine viruses or members of pestiviruses were firstly tested by nPCR, except for ASFV that was used certified reference genomic nucleic acid material as template and tested by OIE recommended PCR"*.

Line 344-347: *"Before comparison of these three methods for testing tissue samples, cDNAs of 50 spleen samples were firstly tested by PCR with individual specific primers of CSFV, PRRSV, PEDV, PRV and PCV2 to investigate the background of each samples with suspected clinical signs"* was changed to *"Before comparison of these three methods for testing tissue samples, cDNAs of 50 spleen samples were firstly tested by PCR with individual specific primers of BVDV, APPV, PRRSV, PEDV, ASFV, PRV and PCV2 to investigate the background of each sample"*.

Line 346-347: *"Results showed a complex infection situations among these samples"* was changed to *"Results showed the infection situations were complex among these samples"*.

Reviewer #2 (Comments for the Author):

In this study, the authors established a method for the differentiation of Classical swine fever virus virulent and vaccine strains by using HUDSON-RT-RAA-CRISPR/Cas13a, the sensitivity and specificity were also evaluated. However, the present assay established need be further evaluated. As a CRISPR detection method, the sensitivity in this assay just only reaches 300 copies/ μ L. At present, I have checked many publications on CRISPR detection methods, and the detection limit can be reached 10 copies or even 1 copy per μ L. Additionally, the comparative experiment with ordinary PCR method is done in this assay, and the sensitivity is the same each other, which causes that this assay has no practical significance in the field, and the sensitivity is not up to that of qPCR. Therefore, I suggest further optimization of the method.

Re: Many thanks. We appreciate the reviewer's carefulness and insightful comments. We have highlighted **changed contents** in green.

As is mentioned in the manuscript, the main disadvantage of CRISPR is poor detecting sensitivity. Thus, pre-amplification of tested samples by PCR and isothermal amplification techniques is the main strategy for sensitivity enhancement. Currently, there is no fixed rules to estimate the performance of primers used for RPA or RAA. Development of RAA with high sensitivity solely rely on large-scale work for candidate primers screening. Considering the main purpose of our work is to differentiate CSFV virulent and vaccine strains, our detecting target focused on the area of 12 nt-insertion at 3'UTR, which limits the number of candidate primers to optimize. Once received comments of Reviewer #2, we screened candidate primers which meet the principles of RAA primer designing around 12 nt-insertion and all candidate primer pairs showed similar performance compared with initial primer pairs. Added contents have been presented in supplementary materials as **figure S1 and table S1**.

Though sensitivity of established CRISPR/Cas13a in this study is the same as nPCR, there is still practical significance in the field. Eradication of CSF need the use of efficient vaccine and DIVA diagnosis. One way is to develop novel marker vaccines such as gene-deleted or subunit ones, while the other is to develop novel diagnostic method that is able to precisely differentiate between virulent and vaccine strains. Considering HCLV based vaccine is one of the safest and most efficient vaccines, development of DIVA diagnostic method is easier and more practical. Due

to the limited sensitivity, immunoassays such as ELISA usually cannot meet the demand for antigen detection. Current studies about CSFV DIVA mainly rely on molecular tests, including sequencing, PCR and restrictive fragment length polymorphism. However, these methods are often laborious and time consuming that need to be performed by skilled technicians with expensive instruments, multiple testing steps and complex reagents. Compared with these methods, the established CRISPR/Cas13a in this manuscript indeed provided a better choice for fast and convenient DIVA of CSFV.

Other points as follow:

1. In Figure 1, the detection method is divided into 3 steps. It is whether the reagent is added by opening the lid according to the steps? If it is added in steps, it is concerned about the possible nucleic acid aerosol contamination after opening the lid between the second and third steps.

Re: We are thankful to the reviewer's insightful comments. Cross-contamination of templates or amplification products is one of the most common problems seen among most isothermal amplification techniques. The established CRISPR/Cas13a used RAA for pre-amplification of tested sample, thus there was indeed a risk of cross-contamination. We concluded and followed some rules to effectively minimize the chances of cross-contamination: (1) Aliquot all experiment materials in small volumes for storage. Defrost and use up all aliquoted materials for one experiment; (2) Use aerosol-resistant pipette tips, avoiding vigorous shaking and pipetting; (3) Properly dispose tips, strips, excess buffer and other related materials after RAA experiments.

2. In Figure 2, the annotation of marker does not match the figure.

Re: We are thankful to the reviewer's insightful comments. The annotation of marker has been modified to match the figure.

3. In Figure 3, the representation of the 3A and 3B is reversed.

Re: We are thankful to the reviewer's insightful comments. The presentation of figure 3 has been corrected.

Changed content in line 568-575: The figure captions were modified to match figure 3.

4. In Figure 9, many non-specific bands can be seen, and I have doubts about the results of this PCR.

Re: We are thankful to the reviewer's insightful comments. In figure 9, nPCR for PEDV of some

samples showed non-specific bands, which may cause confused results. Thus, we optimized the PCR conditions and tested 50 clinical samples. The performance of PCR for PEDV was greatly improved. New results were presented in **figure S2** and **table 5**. Modified PCR conditions were shown in table

5. It is recommended to incorporate some figures and tables to enhance the readability of the manuscript.

Re: We are thankful to the reviewer's insightful comments. We have merged figure figure 9, figure 10 and figure 11 as one new **figure S2** and presented as supplementary materials. Alternatively, we reorganized results of comparison experiment of nPCR, HUDSON-RT-RAA-CRISPR/Cas13a and antigen ELISA for testing 50 clinical samples and presented as **table 5**.

6. Line 309, cNDA should be modified to cDNA; Line 311 , PCV 2 should remove the space.

Re: We are thankful to the reviewer's insightful comments. We have corrected related contents throughout the manuscript.

Responses to reviewer #3 comments:

In the present study, Zhang et al. reports a CRISPR/Cas13a based diagnostic method for differential detection of virulent and vaccine strains of classical swine fever virus (CSFV). Considering CSF eradication is undergoing in China and countries that use hog cholera lapinized virus (HCLV) derived live attenuated live vaccines, this method has important research significance for it provides a promising strategy for CSF diagnosis and eradication. However, there are still some major and minor concerns that need author to be improved as following:

Re: Many thanks. We have exhibited **changed contents** with red color according to reviewer #3 comments in the revised manuscript.

Major concerns:

1. Line 33, " In combination with recombinase-aided amplification (RAA) ". As it is known, there are currently two recombinase-based amplification techniques, including recombinase polymerase amplification (RPA) and recombinase-aided amplification (RAA), of which RPA is the most popular amplification technique that combined with CRISPR. What are the differences between RPA and RAA? Is there any reason that RAA is chose in the present manuscript?

Re: We are thankful to the reviewer's insightful comments. Currently, there are two kinds of recombinase-based amplification techniques, which are recombinase polymerase amplification (RPA) and recombinase-aided amplification (RAA). In general, RPA and RAA share the same steps and mechanism for nucleic acid amplification. The only difference between RPA and RAA is the source of recombinase used. The recombinase used in RPA reaction is from T4 phage, while the recombinase used in RAA reaction is from *E.coli*.

Though RPA is the first amplification technique combined with CRISPR technique, there are increasing number of studies using RAA to improve detecting sensitivity as well. More importantly, in a recent paper about rapid detection for African swine fever virus (ASFV), RPA and RAA were compared in parallel. Results showed that RAA was about two-fold more sensitive than RPA for ASFV detection in that paper. Thus, we finally chose RAA in our study.

Added content in Line 300-303: *A recent study compared two recombinase-based technique, RPA and RAA in parallel for rapid detection for African swine fever virus (ASFV), demonstrating*

the detection limit for RPA was 93.4 copies per reaction while the detection limit for RPA was 53.6 copies per reaction. Thus, RAA was introduced to improve detecting sensitivity in this paper.

2. Line 115, " 50 spleen tissues collected from pigs with suspected clinical signs". The clinical signs should be described more specifically. In other words, what signs that are likely to be caused by CSFV and may confused with other porcine virus infection.

Re: We are thankful to the reviewer's insightful comments. The specific clinical signs were clarified according to diagnostic records of tested samples. In general, we selected spleen samples from pigs showed one or more symptoms such as pyrexia, huddling, weakness, conjunctivitis or diarrhea.

Changed content in line 137-139: *Fifty spleen tissues collected during 2016-2018 from pigs with one or more suspected clinical signs including pyrexia, huddling, weakness, conjunctivitis and diarrhoea.*

3. Line 163, " Viral cell cultures or tissue honogenates (100 mg sample in 500 μ L PBS)". It is confused to understand how tissue honogenates were prepared, a more detailed description is needed to be clarified.

Re: We are thankful to the reviewer's insightful comments. We have modified the related content according to reviewer's instruction.

Changed content in line 186-188: *Spleen tissue honogenates were prepared with a grinder in a ratio of 1:5 (tissue weight/PBS volume, mg/ μ L). 500 μ L viral cell cultures or spleen tissue honogenates were freeze-thawed 3 times and centrifuged at 12000g for 20 min. All procedures were performed at 4 °C.*

4. Line 219, figure 8 and figure 11, " results were calculated according to the manufacturer's instruction". There is a need to introduce how to decide the ELISA results. Similarly, how to determine the positive and negative lines in figure 8 and figure 11?

Re: We are thankful to the reviewer's insightful comments. Specific rules to determine positive and negative results have been added to the manuscript.

Added content in line 242-249: *In brief, OD_{450nm} values of negative control (N), positive control (P) and tested samples (S) were measured with POLARstar Omega multi-function reader (BIO-GENE Biotech, China). For a reliable experiment, N value should be less than 0.250, while the value of (P-N) should more than 0.150. Results were then calculated according to the*

following rules: Positive result was determined if the value of (S-N) was more than 0.300; Suspected result was determined if the value of (S-N) was between 0.100 and 0.300. Negative result was determined if the value of (S-N) was less than 0.100.

Minor concerns:

1. There are some abbreviations such as CSFV, HUDSON, etc., should be given the full name when they appeared for the first time.

Re: We are thankful to the reviewer's insightful comments. Full names of all abbreviations have been given throughout this manuscript.

Modified content in line 48-49: *In this study, a diagnostic platform based on CRISPR/Cas13a was established, with the ability to differentiate between classical swine fever virus (CSFV) virulent and vaccine strains.*

Modified content in line 50-51: *In combination with reverse-transcription recombinase-aided amplification (RT-RAA).*

Modified content in line 71-72: *Classical swine fever (CSF) is a highly contagious and often fatal infectious porcine disease.*

Modified content in line 73-75: *It has brought great economic losses to the swine industry worldwide during the past decades and thus classified as one of notifiable terrestrial and aquatic animal diseases by World Organization for Animal Health (OIE).*

Modified content in line 82-84: *Since the effective and safe attenuated live vaccine hog cholera lapinized virus (HCLV) strain was produced by serial passage of CSFV Shimen strain in rabbits at 1956.*

Modified content in line 103-104: *Cas13a cleaves both the target RNAs as well as non-specific single stranded RNA (ssRNA).*

Modified content in line 112-113: *HUDSON (Heating Unextracted Diagnostic Samples to Obliterate Nucleases) treatment can lyse viral particles and inactivate ribonucleases with the use of heat and chemical reduction.*

Modified content in line 118-119: *Combined the high sensitivity of recombinase-aided amplification (RAA) with the accurate molecular sensing ability of Cas13a.*

Modified content in line 218-219: *CSFV nested PCR (nPCR) was performed according to OIE recommend and previous study with some modifications.*

2. Line 110, "synthesized by Sangon Biotech". Please indicate the country of the company. In addition, all reagents and company in this manuscript should also indicate country.

Re: We are thankful to the reviewer's insightful comments. Countries of all reagents and companies have been indicated throughout this manuscript.

Modified content in line 131-132: *Other DNA oligos, including primers and crRNA transcription templates were also synthesized by Sangon Biotech, China.*

Modified content in line 200-201: *murine RNase inhibitor (NEB, USA) 1.6 unit/ μ L.*

June 18, 2022

Dr. Gaiping Zhang
Key Laboratory of Animal Immunology, Henan Academy of Agricultural Sciences
No.116 Huayuan Road
Zhengzhou
China

Re: Spectrum00891-22R1 (Differentiation of Classical Swine Fever Virus Virulent and Vaccine Strains by CRISPR/Cas13a)

Dear Dr. Gaiping Zhang:

Thank you for submitting your manuscript to Microbiology Spectrum. As you will see, your manuscript is very close to be accepted. But could you please cut down these illustrations further? My advice is merging some related figures into one, or putting over-sized figure(s)/table(s) into supplemental information.

Link Not Available

Sincerely,

Biao He

Journals Department
Reviewer comments:

Staff Comments:

Preparing Revision Guidelines

Please return the manuscript within 60 days; if you cannot complete the modification within this time period, please contact me. If you do not wish to modify the manuscript and prefer to submit it to another journal, please notify me of your decision immediately so that the manuscript may be formally withdrawn from consideration by Microbiology Spectrum.

Revision Notes on manuscript “Spectrum00891-22R1”

Dear Editor:

Thank you for your kind letter of “Spectrum00891-22R1” on June 18, 2022. Generally, we appreciate the editor’s insightful comments, which were helpful for improving the manuscript. Based on your comments and requests, which is “merging some related figures into one, or putting over-sized figure(s)/table(s) into supplemental information”, we have made modifications in the manuscript Spectrum00891-22R1 as following and **highlighted changes in blue** in the revised one:

1. We removed **Figure 2** (SDS-PAGE of LwaCas13a purification) in the manuscript Spectrum00891-22R1 and presented it as **Figure S1** in supplemental information;
2. We removed **Figure 3** (CSFV candidate crRNAs screening by LwaCas13a collateral cleavage assay) in the manuscript Spectrum00891-22R1 and presented it as **Figure S2** in supplemental information;
3. We renumbered **Figure 4** (Estimation of the ability of CRISPR/Cas13a to recognize CSFV different genotypes with screened crRNAs) in the manuscript Spectrum00891-22R1 as **Figure 2** in the revised manuscript;
4. We renumbered **Figure 5** (Detecting sensitivity of CRISPR/Cas13a for synthetic CSFV 3'UTR RNA) in the manuscript Spectrum00891-22R1 as **Figure 3** in the revised manuscript;
5. We renumbered **Figure 6** (Specificity of RT-RAA-CRISPR/Cas13a) in the manuscript Spectrum00891-22R1 as **Figure 4** in the revised manuscript;
6. We renumbered **Figure 7** (Estimation of HUDSON-RT-RAA-CRISPR/Cas13a for direct CSFV cell culture detection) in the manuscript Spectrum00891-22R1 as **Figure 5** in the revised manuscript;
7. We renumbered **Figure 8** (Comparison of HUDSON-RT-RAA-CRISPR/Cas13a, nPCR and

antigen ELISA in detecting 10-fold dilutions of CSFV cell cultures) in the manuscript Spectrum00891-22R1 as **Figure 6** in the revised manuscript;

8. We renumbered **Figure S1** (Design and screening of candidate CSFV RAA primers) in the original supplemental information file as **Figure S3** in the new supplemental information file;

9. We renumbered **Figure S2** (Comparison of nPCR, HUDSON-RT-RAA-CRISPR/Cas13a and antigen ELISA for testing 50 spleen tissue samples with suspected signs) in the original supplemental information file as **Figure S4** in the new supplemental information file;

10. We removed **table 1** (CSFV reference strains used in this study) in the manuscript Spectrum00891-22R1 and presented it as **table S1** in supplemental information;

11. We removed **table 2** (Viruses used in this study) in the manuscript Spectrum00891-22R1 and presented it as **table S2** in supplemental information;

12. We removed **table 3** (crRNAs used in this study) in the manuscript Spectrum00891-22R1 and presented it as **table S3** in supplemental information;

13. We removed **table 4** (Primers and conditions of PCR used in this study) in the manuscript Spectrum00891-22R1 and presented it as **table S4** in supplemental information;

14. We renumbered **table S1** (Candidate CSFV RAA primer pairs) in the original supplemental information file as **table S5** in the new supplemental information file;

15. We removed **table 5** (Results of nPCR, HUDSON-RT-RAA-CRISPR/Cas13a and ELISA for testing fifty spleen tissues) in the manuscript Spectrum00891-22R1 and presented it as **table S6** in supplemental information;

16. We added a new table in revised manuscript as **table 1** (Comparison of HUDSON-RT-RAA-CRISPR/Cas13a, nested PCR and antigen ELISA in testing fifty spleen samples).

Thank you again for your comments. Besides modifications of the revise manuscript, we also

have some issues to communicate with editor:

1. There is a mistake of second co-author. During the first vision of the manuscript, the name of second co-author is “Qimei Li” on the website of the journal, while the correct name is “Qingmei Li”. We failed to correct the name online, and we are hopeful to ask editor’s kind help to correct the name.
2. As listed in the manuscript, there are two corresponding authors in the manuscript. The first corresponding author is Gaiping Zhang, while the second one is Junqing Guo. However, we cannot choose Junqing Guo as corresponding author online. We are hopeful to ask editor’s kind help to choose Junqing Guo as second corresponding author.

July 10, 2022

Dr. Gaiping Zhang
Key Laboratory of Animal Immunology, Henan Academy of Agricultural Sciences
No.116 Huayuan Road
Zhengzhou
China

Re: Spectrum00891-22R2 (Differentiation of Classical Swine Fever Virus Virulent and Vaccine Strains by CRISPR/Cas13a)

Dear Dr. Gaiping Zhang:

Your manuscript has been accepted, and I am forwarding it to the ASM Journals Department for publication. You will be notified when your proofs are ready to be viewed.

Sincerely,

Biao He
Editor, Microbiology Spectrum
